# Neighborhood Beauty and the Brain in Older Japanese Adults

**DOI:** 10.3390/ijerph20010679

**Published:** 2022-12-30

**Authors:** Yukako Tani, Takeo Fujiwara, Genichi Sugihara, Masamichi Hanazato, Norimichi Suzuki, Masaki Machida, Shiho Amagasa, Hiroshi Murayama, Shigeru Inoue, Yugo Shobugawa

**Affiliations:** 1Department of Global Health Promotion, Tokyo Medical and Dental University (TMDU), Tokyo 113-8510, Japan; 2Department of Psychiatry and Behavioral Sciences, Tokyo Medical and Dental University (TMDU), Tokyo 113-8510, Japan; 3Center for Preventive Medical Sciences, Chiba University, Chiba 263-8522, Japan; 4Department of Preventive Medicine and Public Health, Tokyo Medical University, Tokyo 160-8402, Japan; 5Research Team for Social Participation and Community Health, Tokyo Metropolitan Institute of Gerontology, Tokyo 173-0015, Japan; 6Department of Active Ageing (Donated by Tokamachi City, Niigata), Niigata University Graduate School of Medical and Dental Sciences, Niigata 951-8510, Japan

**Keywords:** neighborhood environment, brain imaging, beauty, aesthetics, green space, older adults

## Abstract

People have a preference for, and feel better in, beautiful natural environments. However, there are no epidemiological studies on the association between neighborhood beauty and neuroimaging measures. We aimed to determine association between neighborhood beauty and regional brain volume. Participants were 476 community-dwelling older adults from the Neuron to Environmental Impact across Generations (NEIGE) study. Subjective neighborhood beauty was assessed through participants’ perception of beautiful scenery within 1 km of their home. Objective measures of neighborhood indicators (green spaces, blue spaces, and plant diversity) within 1 km of participants’ homes were obtained using a geographic information system. Volumes of brain regions associated with experience of beauty were measured using magnetic resonance imaging. We estimated associations between neighborhood beauty and regional brain volume using linear regression. Of the participants, 42% rated their neighborhoods as very beautiful, and 17% rated them as not at all beautiful. Higher subjective neighborhood beauty was associated with larger bilateral medial orbitofrontal cortex and insula volumes (all *p* for trend < 0.01). Brain volume was not associated with objective neighborhood measures. Subjective neighborhood beauty was associated with brain regions related to rewards and decision making, suggesting that these brain regions underpin the perception of neighborhood beauty.

## 1. Introduction

The preference for beautiful scenery dates back to ancient Greece [1]. Research shows that exposure to green space is associated with both physical and mental well-being [2,3,4]. However, the effects of exposure to the surrounding environment have not been investigated from an aesthetic point of view. Subjective perspectives on the neighborhood environment may be important in investigations of the effect of neighborhood environment on health. Whether beauty resides in the perceived object or in the perceiving subject is one of the most debated questions in aesthetics; however, it is generally accepted that subjective criteria play a major role in aesthetic perception [5]. Because beauty is a subjective perception, aesthetic reactions can be shaped by the viewer’s experiences, life events, education, values, and health status [6,7]. Therefore, the effect of a particular environment on humans can depend on subjective factors.

In recent decades, neuroimaging research has been conducted to elucidate the neural mechanisms underlying the experience of beauty. Studies have consistently shown that the experience of beauty is associated with the activity of the medial orbitofrontal cortex (mOFC) [5,8,9,10]. For example, studies using functional magnetic resonance imaging (fMRI) show that the mOFC was activated when participants viewed paintings that they considered beautiful [5], experienced beauty in visual and musical domains [8], and made aesthetic judgments [9,10]. The insula has also been linked to the experience of beauty, such as the concept of ideal beauty and perception of the golden ratio [6]. Furthermore, mOFC and insular cortex displayed an opposing relationship during attractiveness and goodness judgments [10]. However, there are no epidemiological studies on the neural effects of exposure to beautiful environments.

Thus, the study aim was to investigate the association between subjective neighborhood beauty and regional brain volume in the mOFC and insula among healthy older people using data from the Neuron to Environmental Impact across Generations (NEIGE) study. The NEIGE was developed to investigate the social determinants of health among older people in rural areas in Japan. Furthermore, we examined whether volumes of the brain regions linked to subjective neighborhood beauty were associated with objective neighborhood beauty, including green spaces, blue spaces, and plant diversity.

## 2. Materials and Methods

### 2.1. Study Design and Participants

We used data from the 2017 NEIGE study, details of which have been described previously [11]. Briefly, a survey was conducted in Tokamachi City, Niigata Prefecture, Japan, among community-dwelling older adults aged 65–84 years, without functional disabilities, and not certified as eligible for long-term care insurance benefits [12]. Tokamachi is an agricultural city with a population density of 87.8 people per km^2^. Study participants (N = 1,346) were randomly selected from four groups stratified by age (65–74 years and 75–84 years) and residential area (Central Tokamachi [downtown area] and Matsunoyama [mountain area]) using the resident register, and invited to participate in the study via mail. A total of 527 people (participation rate: 39.2%) agreed to undergo examination and participate in the study. The analytic sample for the present study comprised 476 participants, after excluding participants receiving treatment for dementia (N = 3), those with a history of psychiatric disease (N = 17), and those with missing brain imaging data (N = 31). The NEIGE protocol was approved by the Human Subjects Committees of Niigata University (No. 2666). Participants were informed that study participation was voluntary, and all participants provided written informed consent. Over 90% of participants had lived in the same location for over 30 years, which enabled us to assess long-term exposure to the neighborhood environment.

### 2.2. Brain Imaging Measures

Brain MRI was conducted at Niigata Prefectural Tokamachi Hospital from 2017 to 2018. Scanning was performed using a 1.5 Tesla scanner (MAGNETOM Avanto fit, Siemens, Germany) used for patients at the hospital [13]. The structural MRI scanning parameters were as follows: repetition time = 1700 ms, echo time = 4.31 ms, flip angle = 15°, field of view = 230 × 230 mm, acquisition matrix size = 256 × 256 mm, slice thickness = 1.25 mm, and number of slices = 144. Scanning time was 15–20 min per person. Structural T1 brain MRI data were acquired and processed using FreeSurfer, version 6.0 [14]. Segmented regions based on the Desikan–Killiany atlas and calculated regional brain volumes for the bilateral mOFC and insula (Figure 1). A manual quality check was conducted. We selected the mOFC and insula regions because neuroimaging studies have shown that the experience of beauty, such as viewing pictures of paintings, is associated with activities in these regions [5,6,8,9,10]. The brain volume data were regressed on intracranial volume and residuals were derived. We obtained the regional brain volume by calculating the sum of the residual and mean volume of each region (Appendix A).

### 2.3. Subjective Neighborhood Beauty

Subjective neighborhood beauty was assessed using a self-report questionnaire. Participants were asked “Are there many attractive landscapes (e.g., beautiful view, enthralling perspective) within walking distance (within 1 km) of your home?” Responses were on a four-point Likert scale ranging from 1 (disagree) to 4 (agree). These responses corresponded to low, moderately low, moderately high, and high subjective neighborhood beauty, respectively.

### 2.4. Objective Neighborhood Indicators (Green Spaces, Blue Spaces, and Plant Diversity)

There is no established method of measuring objective beauty, but it is considered an aspect of beauty that exists outside of cultural trends and personal preferences [15]. Many aspects of nature are intrinsically beautiful and aesthetically pleasing, such as the golden ratio and the Fibonacci sequence [16,17]. Therefore, we assessed several natural indicators to compare with subjective neighborhood beauty. Green spaces, blue spaces, and plant diversity were assessed using remote sensing data. A neighborhood was defined as the 1-km buffer zone around the participants’ home addresses. We used the normalized difference vegetation index (NDVI) from the 30 m resolution Landsat 8 data (USGS, Reston, VI, USA) collected on 31 August 2016 as an index of green space. The NDVI measures the difference between near infrared wavelengths (which vegetation strongly reflects) and red wavelengths (which vegetation absorbs) and ranges from −1 to +1 [18]. A positive NDVI indicates that the land cover is likely to be green vegetation. Blue spaces were assessed using a geographic information system data-generated 1:25,000 scale vegetation map that has been produced since 2005 by the Biodiversity Center of Japan, Ministry of the Environment [19]. These data contain polygons that represent types of vegetation and land use. We calculated two types of blue spaces, one with paddy fields and the other without paddy fields, because the study area contains “tanada” (rice field terraces) that line the mountain slopes; their appearance changes with the seasons, producing different vibrant colors as the year progresses [20]. Plant diversity was calculated using Simpson’s diversity index, which ranges from 0 to 1 [21], based on the geographic information system data-generated vegetation map. The vegetation map shows the distribution of plant species (cedar, cypress, pine, oak, beech, etc.). In addition, it shows whether these plants were planted artificially or not. In Tokamachi City, 52 plant species are registered (Appendix A).

### 2.5. Covariates

Covariates were assessed using self-report questionnaires. Age was categorized into four groups (65–69, 70–74, 75–79, and ≥80 years). Educational attainment was categorized into three groups by years of schooling (≤9, 10–12, and ≥13 years). Annual household income was categorized into three groups (<2.00, 2.00–3.99, and ≥4.00 million yen). Marital status was categorized into four groups (married, widowed, divorced, and not married). The age at which participants first lived in their current location was calculated by subtracting the period of residence from their age and categorizing it into four groups (≤15, 16–24, 25–39, and ≥40 years old). Childhood exposure to neighborhood nature was assessed by the following question: “When you were a child, did you have the opportunity to interact with the immediate natural environment, such as taking a walk in a green area?” Responses were on a five-point Likert scale ranging from 1 (always) to 5 (not at all). These responses corresponded to high, moderately high, medium, moderately low, and low exposure to neighborhood nature during childhood, respectively.

### 2.6. Statistical Analysis

First, associations between participant characteristics and subjective neighborhood beauty were examined using the chi-square test. Second, the associations between neighborhood variables were calculated using Spearman’s correlation coefficient. Third, linear regression models were used to examine the association between subjective neighborhood beauty and regional brain volume. On the basis of previous study findings [5,6,8,9,10], we hypothesized that living in a subjectively beautiful neighborhood would be associated with mOFC and insula volumes. Model 1 was a crude model. Model 2 adjusted for the potential confounders of age, sex, education, and income. Model 3 additionally adjusted for residential area and childhood exposure to neighborhood nature because we wished to test whether there was a direct association between subjective neighborhood beauty and regional brain volume, excluding the effect of residential area and childhood experience. We constructed a directed acyclic graph (DAG) of proposed associations between and to guide our analyses (Appendix A). We further stratified our analyses by residential area (mountain or downtown area) because the effect of neighborhood beauty may differ according to neighborhood size and type. Finally, linear regression models were used to examine the association between objective neighborhood indicators (green spaces, blue spaces, and plant diversity) and brain regions to investigate which brain regions correlated with objectively measured neighborhood variables. All analyses were conducted using Stata, version 15, with the significance level set at 0.05.

## 3. Results

Participant characteristics are shown in Table 1. Of the participants, 48% were men, 18% were ≥80 years old, 37% had <9 years of education, 40% had an annual household income of <2 million yen, and 81% were married. Approximately 40% of participants had lived in their current location since they were children (≤15 years old) and 70% had first lived in their current location before the age of 25 years. Regarding subjective neighborhood beauty, 42% of participants rated their neighborhood as very beautiful (high), whereas 17% rated their neighborhood as not at all beautiful (low). Most participant characteristics, including age, sex, sociodemographic variables, and age of first living in the current location, were not associated with subjective neighborhood beauty. Older adults living in Matsunoyama (a mountain area) rated their neighborhood environment as more beautiful than those living in Central Tokamachi (a downtown area). Older people with greater exposure to nature as children rated their current neighborhood environment as beautiful.

Subjective neighborhood beauty was significantly and positively correlated with objectively measured green spaces (r = 0.31, *p* < 0.001); blue spaces, including paddy fields; (r = 0.28, *p* < 0.001); and plant diversity (r = 0.29, *p* < 0.001), but negatively correlated with objectively measured blue spaces that did not include paddy fields (r = −0.13, *p* = 0.006) (Table 2). Objectively measured green space was highly positively correlated with plant diversity.

The associations between subjective neighborhood beauty and regional brain volume are shown in Table 3. Older adults with high subjective perceptions of neighborhood beauty had larger bilateral mOFC (left: *p* for trend = 0.001, right: *p* for trend = 0.01) and bilateral insula (left: *p* for trend = 0.002, right: *p* for trend < 0.0001) volumes than those with low perceptions of subjective neighborhood beauty. These associations remained significant after adjusting for individual-level variables (age, sex, education, and income) (Model 2). After adjusting for residential area and childhood exposure to neighborhood nature, these associations were lower but remained significant (Model 3). Analyses stratified by residential area showed that the association between subjective neighborhood beauty and regional brain volume tended to be stronger in participants who lived in the mountainous area than in participants who lived in the downtown area (Appendix A).

The associations between objective neighborhood indicators and regional brain volume are shown in Table 4. The linear regression analysis showed that objectively measured neighborhood green spaces, blue spaces, and plant diversity were not associated with mOFC and insula volumes (all *p* > 0.08).

## 4. Discussion

To our knowledge, this is the first epidemiological study to examine the associations between exposure to beautiful environments and neuroimaging measures. We found that older people with high subjective neighborhood beauty ratings had larger mOFC and insula volumes than those with low subjective neighborhood beauty ratings. We also found that the volumes of these brain regions were not associated with objective neighborhood indicators including green spaces, blue spaces, and plant diversity.

Subjective neighborhood beauty was positively associated with mOFC volume. This finding is in line with the results of fMRI studies of the experience of beauty [5,6,8,9,10]. Neuroimaging studies have shown that the orbitofrontal cortex is a heterogeneous brain region with many functions, such as sensory integration, modulation of visceral reactions, and decision making in emotional and reward-related behaviors [22,23]. In particular, the human orbitofrontal cortex has been linked to the subjective experience of pleasantness [22,23]. A quantitative meta-analysis showed that subjective pleasantness ratings (mostly ratings of pleasantness, attractiveness, or beauty) were associated with mOFC activation [24]. Because most people are exposed to their neighborhood environment in daily life, living in a beautiful environment may activate the mOFC, which leads to an increase in mOFC volume.

Subjective neighborhood beauty was also positively associated with insula volume, which is in line with previous findings on the experience of beauty [6,9]. The insula plays a fundamental role in human emotional awareness and interoception [25,26]. In addition, insular volume has been associated with well-being (personal growth, positive relations, and purpose in life) [27]. Therefore, a large insula volume may be an indicator of greater self-awareness and a more developed sense of beauty.

Although this study employed structural measures of MRI, and thus, caution should be exercised when referring to results of functional MRI studies, structural measures (e.g., volume and thickness) in a brain region have been linked to function in the region [28,29]. In general, regional brain volume decreases with age [30]. Furthermore, brain imaging studies of dementia and psychiatric disorders have shown that a decrease in regional brain volume is associated with functional decline in that region [31]. In our study, older people with high subjective neighborhood beauty ratings had larger mOFC and insula volumes. In addition to the functions described above, both mOFC and insula play an important role in value-based decision-making [23]. These functions have been reported to be impaired in psychiatric disorders [32]. Therefore, the greater volume of mOFC and insula volume may preserve these functions and contribute to the well-being of older adults.

Objectively measured neighborhood green spaces, blue spaces, and plant diversity were not associated with mOFC and insula volumes. One possible explanation is that these brain regions are involved in only subjective or emotional experiences and perceptions of beauty. In this study, objective neighborhood variables were measured using several natural indicators. This was based on the fact that nature often contains intrinsically beautiful characteristics, such as the golden ratio and the Fibonacci sequence, which are aesthetically pleasing [16,17]. However, such objective beauty does not take into account individual preferences [15]. For example, some people prefer rural neighborhoods rich in nature, whereas others prefer urban neighborhoods that feature less nature. Such preferences may explain the discrepancy we identified in the associations of subjective beauty and objective variables with brain volume.

Objective neighborhood variables may be associated with brain regions that were not investigated in this study. For instance, a higher prefrontal cortex and premotor cortex volume in children was associated with neighborhood green spaces, as assessed by the NDVI [33]. One study identified an association between reduced subgenual prefrontal cortex activity and walking in a natural environment compared with walking in an urban environment [34]. Another possible explanation for the lack of association between objective measures and brain volume is the distribution of the objective neighborhood variables. That the study area is completely surrounded by a rich natural environment may have hindered the detection of an association between objective neighborhood variables and neuroimaging measures.

Another reason for the discrepancy between subjective beauty and objective variables findings is that subjective beauty measures may capture non-natural factors such as landmarks. There are many shrines, museums, and community centers in our survey area where festivals and local events are held [35]. Japanese people appreciate the beauty of highly functional utensils, as expressed in the Mingei or folk-craft movement [36]. Subjective beauty measures may reflect the beauty of local cultures. This view is supported by the finding that subjective neighborhood beauty was weakly correlated with objectively measured neighborhood green and blue spaces (all correlations were approximately 0.3). We found that childhood exposure to neighborhood nature was associated with subjective neighborhood beauty, indicating the importance of personal experiences to subjective beauty [6,7]. Further research is needed on the determinants of subjective beauty.

This study had several limitations. First, subjective neighborhood beauty was assessed using a single-item scale, which has not been validated. However, we confirmed that participants who lived in Matsunoyama, which has many hot springs, is rich in nature, and is a popular tourist destination [35], rated their neighborhood as more beautiful than those who lived downtown. This suggests that the scale has some validity. However, this scale should be expanded by adding more items that measure subjective beauty, and its validity and reliability tested. Second, we could not assess causality because this was a cross-sectional study; we can only conclude that individuals with larger mOFC and insula volumes were more likely to report subjective neighborhood beauty. However, most participants had lived in the same location for over 30 years (since they were young), which suggests that accumulated neighborhood exposure may contribute to the development of brain volume in specific regions. Additionally, we excluded participants with a history of psychiatric disorders, which are associated with brain volume [37,38,39,40,41]. Finally, it is difficult to generalize the findings to older adults in other areas because the subjective perception of neighborhood beauty may vary with culture.

## 5. Conclusions

We identified several brain regions associated with living in a subjectively beautiful environment. We were able to link aesthetic exposure to specific phenotypes using objectively measurable brain images, which adds quantitative data to the qualitative research findings in this area. Older people with high subjective ratings of neighborhood beauty had larger mOFC and insula volumes than those with low subjective ratings of neighborhood beauty. These brain regions were not associated with exposure to objective neighborhood indicators such as green spaces, blue spaces, and plant diversity. Future studies should investigate whether this association holds for other populations and other regions. Clarification of the association between neighborhood beauty and health would contribute to health-friendly city design.

## Figures and Tables

**Figure 1 ijerph-20-00679-f001:**
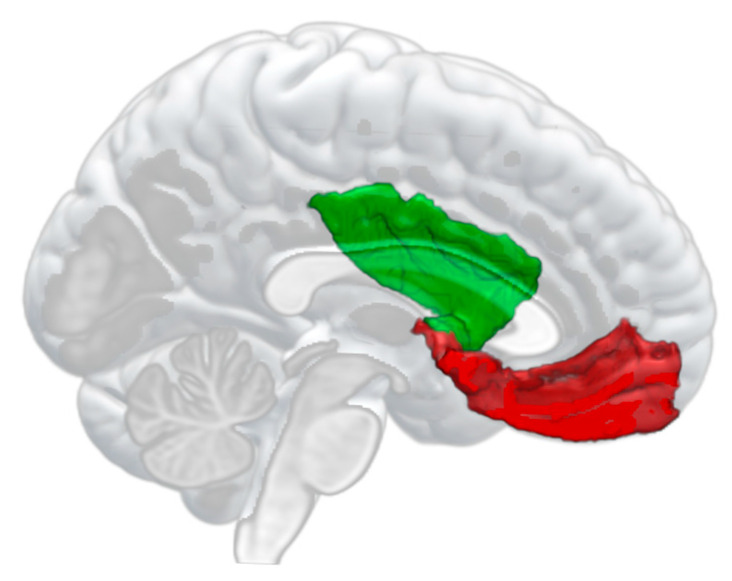
Example of regions of interest in the left hemisphere. Red, medial orbitofrontal cortex; green, insula.

**Table 1 ijerph-20-00679-t001:** Characteristics of participants (n = 476).

	Total		Subjective Environmental Beauty				
			Low	Moderately Low	Moderately High	High	*p* Value ^a^
	n = 476		n = 81	n = 95	n = 100	n = 200	
	n	%	%	%	%	%	
Age (years)							
65–69	153	32.1	23.5	38.9	34.0	31.5	0.45
70–74	129	27.1	25.9	29.5	23.0	28.5	
75–79	109	22.9	28.4	16.8	23.0	23.5	
≥80	85	17.9	22.2	14.7	20.0	16.5	
Sex							
Male	230	48.3	50.6	44.2	53.0	47.0	0.61
Female	246	51.7	49.4	55.8	47.0	53.0	
Education (years)							
Low (≤9)	178	37.4	38.3	27.4	34.0	43.5	0.13
Moderate (10–12)	203	42.6	37.0	49.5	45.0	40.5	
High (≥13)	95	20.0	24.7	23.2	21.0	16.0	
Annual income (million yen)							
Low (<2.00)	190	39.9	44.4	34.7	35.0	43.0	0.06
Moderate (2.00–3.99)	206	43.3	42.0	44.2	46.0	42.0	
High (≥4.00)	51	10.7	3.7	17.9	10.0	10.5	
Missing	29	6.1	9.9	3.2	9.0	4.5	
Marital status							
Married	387	81.3	82.7	82.1	74.0	84.0	0.70
Widowed	73	15.3	13.6	14.7	20.0	14.0	
Divorced	9	1.9	2.5	2.1	3.0	1.0	
Not married	7	1.5	1.2	1.1	3.0	1.0	
Age first lived in current location							
≤15 years old	182	38.2	43.2	33.7	39.0	38.0	0.09
16–24 years old	153	32.1	23.5	31.6	28.0	38.0	
25–39 years old	93	19.5	19.8	25.3	26.0	13.5	
≥40 years old	48	10.1	13.6	9.5	7.0	10.5	
Residential area							
Matsunoyama (mountain)	174	36.6	25.9	15.8	30.0	54.0	<0.001
Central Tokamachi (downtown)	302	63.4	74.1	84.2	70.0	46.0	
Childhood exposure to neighborhood nature							
High	389	81.7	71.6	66.3	85.0	91.5	<0.001
Moderately high	45	9.5	4.9	17.9	12.0	6.0	
Medium	18	3.8	7.4	5.3	2.0	2.5	
Moderately low	18	3.8	12.3	7.4	1.0	0.0	
Low	6	1.3	3.7	3.2	0.0	0.0	

^a^ Differences were tested using Pearson’s chi-square test.

**Table 2 ijerph-20-00679-t002:** Summary of neighborhood measures and their Spearman correlations (n = 476).

	Neighborhood Variables	Mean	SD	Median	Min	Max	1	2	3	4	5
1	Subjective beauty (range: 1–4)	2.88	1.14	3	1	4	1.00				
2	Objective green space (NDVI, range: −1 to 1)	0.34	0.11	0.37	0.19	0.48	**0.31**	1.00			
3	Objective blue space (km^2^)	0.029	0.038	0.024	0	0.32	**−0.13**	**−0.33**	1.00		
4	Objective blue space including paddy fields (km^2^)	0.65	0.43	0.62	0	1.94	**0.28**	**0.59**	−0.05	1.00	
5	Objective plant diversity (range: 0–1)	0.60	0.18	0.67	0.31	0.85	**0.29**	**0.83**	**−0.29**	**0.61**	1.00

Boldface indicates statistical significance (*p* < 0.05). NDVI = normalized difference vegetation index; SD = standard deviation.

**Table 3 ijerph-20-00679-t003:** Associations between subjective neighborhood beauty and regional brain volume among Japanese older adults (n = 476).

	mOFC (mm^3^)		Insula (mm^3^)	
	Left	Right	Left	Right
	Coef. (95% CI)	Coef. (95% CI)	Coef. (95% CI)	Coef. (95% CI)
Model 1				
Subjective beauty				
Low	referent	referent	referent	referent
Middle-low	**211 (69 to 354)**	129 (−8.1 to 266)	127 (−32 to 286)	173 (−4.9 to 352)
Middle-high	**232 (91 to 373)**	**143 (7.7 to 279)**	**222 (65 to 379)**	**253 (76 to 429)**
High	**242 (118 to 366)**	**166 (47 to 286)**	**218 (79 to 356)**	**317 (162 to 472)**
*p* for trend	0.001	0.01	0.002	<0.0001
Model 2				
Subjective beauty				
Low	referent	referent	referent	referent
Middle-low	**155 (14 to 296)**	89 (−50 to 227)	110 (−53 to 273)	118 (−63 to 298)
Middle-high	**205 (67 to 342)**	134 (−1.2 to 268)	**219 (60 to 378)**	**227 (51 to 403)**
High	**204 (82 to 326)**	**148 (28 to 268)**	**210 (69 to 351)**	**294 (137 to 450)**
*p* for trend	0.003	0.02	0.003	<0.0001
Model 3				
Subjective beauty				
Low	referent	referent	referent	referent
Middle-low	**158 (15 to 301)**	90 (−50 to 230)	112 (−54 to 277)	125 (−58 to 308)
Middle-high	**207 (66 to 349)**	117 (−21 to 255)	**215 (52 to 379)**	**232 (51 to 413)**
High	**202 (73 to 331)**	**126 (0.4 to 252)**	**203 (54 to 352)**	**295 (130 to 460)**
*p* for trend	0.005	0.07	0.007	<0.0001

Coef.: Regression coefficients; CI = confidence interval; mOFC = medial orbitofrontal cortex. Model 1: Crude model. Model 2: Adjusted for age, sex, education, and income. Model 3: Model 2 + residential area and childhood exposure to neighborhood nature. Boldface indicates statistical significance (*p* < 0.05).

**Table 4 ijerph-20-00679-t004:** Associations between objective neighborhood indicators and brain volume among Japanese older adults (n = 476).

Objective Neighborhood Variables	mOFC (mm^3^)		Insula (mm^3^)	
	Left	Right	Left	Right
	Coef. (95% CI)	Coef. (95% CI)	Coef. (95% CI)	Coef. (95% CI)
Model 1				
Green space (NDVI, −1 to 1)	37 (−349 to 424)	203 (−165 to 572)	211 (−217 to 640)	264 (−220 to 747)
*p* value	0.85	0.28	0.33	0.28
Blue space (km^2^)	−561 (−1712 to 590)	−986 (−2081 to 110)	−44 (−1321 to 1234)	−644 (−2084 to 797)
*p* value	0.34	0.08	0.95	0.38
Blue space including paddy fields (km^2^)	−55 (−157 to 46)	−30 (−126 to 67)	74 (−38 to 186)	58 (−68 to 185)
*p* value	0.28	0.55	0.20	0.37
Species diversity index (0–1)	47 (−203 to 297)	105 (−134 to 344)	95 (−184 to 373)	99 (−214 to 413)
*p* value	0.71	0.39	0.50	0.53

Coef.: Regression coefficients; CI = confidence interval; mOFC = medial orbitofrontal cortex; NDVI = normalized difference vegetation index. Model 1: Crude model.

## Data Availability

The datasets generated and analysed during the current study are not publicly available due ethical or legal restrictions but are available from the corresponding author on reasonable request.

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
