# Peer review of "Neighborhood Beauty and the Brain in Older Japanese Adults"

_ijerph, 2022, doi:10.3390/ijerph20010679_

Round 1

Reviewer 1 Report

Overall, this paper is well written with clear structure, focused content, and succinct and plain English. The conclusion is supported by the data and analysis.

I think the authors can improve the paper by better presentation of the research context and information and enrichment of discussion part. Following are comments:

1. The main finding is that older adults with high subjective perceptions of neighborhood beauty had larger bilateral mOFC and bilateral insula volumes than those with low perceptions of subjective neighborhood beauty. I think authors need to provide more discussions on such finding and its implications. What is the benefit to have larger bilateral mOFC and bilateral insula volume? What does it mean in health and welbeing? Such discussion will enrich the implication and conclusion of the research. Even though there is some discussion on it but not enough.

2. No figure was provided on the observed part of the brain. For example, providing figures showing the location of the bilateral mOFC and bilateral insula in brain will be informative to readers. It would be informative as well to provide example MRI images showing differences in brain of high subjective perception of neighborhood beauty and low.

3. There is not enough information on the research setting. How long did it take for MRI scanning of one participant? Photos of experiment room, scanner, etc would be good. Even though authors mentioned that the detail was reported in their previous paper, I think this manuscript itself should deliver the research context enough for clear understanding.

Reviewer 2 Report

I absolutely disagree that green spaces, blue spaces, and plant diversity can be named „neighbourhood beauty” measures. This is conceptually wrong and I would rephrase throughout the manuscript.

What scanning modality was used? T1? Which atlas was used to extract information on brain regions’ volume? These essential details are missing.

Definitely, the diversity data should be described in more detail. It seems to rather be land diversity data than plant diversity, too. As such, naming is not correct.

What motivated covariate selection, especially, childhood exposure to nature? DAG would be helpful. It seems to me that the latter better qualifies as a mediator.

In what units brain volume of the regions were measured? This would help to understand regression results better.

I am not sure it is correct to compare results from sMRI study to fMRI studies.
